# Attachment and epistemic trust in junior and senior university students: The mediating role of affect regulation and mentalizing. Who is at risk?

Evangelia Karagiannopoulou[1,2]*, Panagiotis Lianos[1]*, Panoraia Andriopoulou[1], Christos Rentzios[1,3], Peter Fonagy[2,4]*

1 Department of Psychology, University of Ioannina, Greece, 2 Research Department of Clinical, Educational and Health Psychology, University College London, London, United Kingdom, 3 Department of Psychology, Neapolis University Pafos, Cyprus, 4 Anna Freud National Centre for Children and Families, London, United Kingdom

* ekaragia@uoi.gr (EK); plianos@uoi.gr (PL); p.fonagy@ucl.ac.uk (PF)

## Abstract

Research on emotional factors and mental health in higher education has gained traction. Much attention has focused on first-year students as a potentially at-risk group, though some studies suggest that all students might face similar risks. This study examines differences between junior and senior undergraduates in terms of mentalizing, emotion regulation (ER), and psychological mindedness, involving cognitive capacities significantly developed by late adolescence. These constructs relate to understanding one's own and others' mental states, potentially mediating the relationship between attachment and epistemic trust (ET). The current study includes 460 undergraduate students, most of whom are female (96%). Results show that senior students score higher on reappraisal, certainty, and interest/curiosity compared to junior students. However, these factors did not mediate the relationship between anxious attachment orientation and ET. Certainty and interest/curiosity mediated the relationship between avoidant attachment orientation and ET, suggesting similar mediation patterns for junior and senior students. On the other hand, suppression and uncertainty/confusion were critical mediators in the relationship between insecure (anxious and avoidant) attachment orientations and epistemic trust. Findings suggest that universities should (a) foster environments that support psychological capacity and facilitate positive learning experiences, and (b) enhance epistemic trust through safe curiosity and develop protective and preventive interventions.

## Introduction

Emotional factors in higher education have garnered considerable attention across various contexts. An increasing number of studies focus on academic emotions, learning, and psycho-social constructs, including those related to well-being and

**Data availability statement:** All relevant data are available at https://reshare.ukdata-service.ac.uk/857480/ (DOI: 10.5255/UKDA-SN-857480).

**Funding:** The author(s) received no specific funding for this work.

**Competing interests:** The authors have declared that no competing interests exist.

mental health [1,2] One area of research examines learning in relation to psychological variables tied to mental health within community samples [3]. The interaction between cognitive and emotional factors is widely recognized, indicating that a significant portion of first- and second-year students exhibit suboptimal learning profiles [4,5]. While these junior students are frequently classified as high risk [6,7] there is limited research on their psychological distinctions from senior students, especially in aspects such as mentalizing and epistemic trust.

Our interest in comparing junior (first and second-year) and senior (third and fourth-year) students is influenced by two areas of research: (a) neuroscience studies suggesting developmental differences in cognitive emotion regulation (ER; reappraisal and suppression) and mentalizing between individuals aged 17–18 and those over 20 [8], aligning with junior and senior student groups, respectively; and (b) educational and mental health literature that identifies first and second-year students as being particularly at risk [9]. Additionally, university students, regardless of their year of study, are often reported to be at risk for mental health issues and poor well-being [10].

Fonagy et al. suggest that epistemic trust, rooted in attachment and developed through mentalizing, is a crucial psychological factor for effective learning [11]. Trusting relationships create an "epistemic superhighway" for learning, reducing the usual epistemic vigilance evolved to handle possible misinformation from others [12]. Given the importance of mentalizing for understanding mental states, this study focuses on psychological mindedness, which involves grasping one's own and others' mental states to interpret behavior. Emotion regulation (ER) also entails representing the mental states of oneself and others, either by paying attention to one's own emotional state or by reevaluating those of others during the reappraisal process. This common ground among mentalizing, psychological mindedness, and emotion regulation places this study within the broader field of affect regulation.

The current research examines how mentalizing, psychological mindedness, and emotion regulation influence the relationship between attachment and epistemic trust. This aligns with studies on university student learning that emphasize individual factors that are hard to change [13] compared to environmental factors related to success. Research indicates that the attachment system plays a key role in the transition to young adulthood during university, with the change resembling a "strange situation" scenario [14].

## Attachment and mentalizing

Mentalizing refers to the capacity to interpret or understand behavior—both one's own and that of others—as psychologically motivated in terms of underlying intentions and mental states, such as thoughts, feelings, wishes, and intentions. It operates across four dimensions: (a) automatic-controlled, (b) internally-externally focused, (c) self-other, and (d) cognitive-affective. This capacity encompasses a wide range of social cognitive processes related to mental states, including perception, recognition, and description [15,16,8].

The roots of mentalizing begin in infancy within early attachment relationships, where the child's complex emotions are "mirrored" by attachment figures in a contingent and marked way [8]. These mirroring interactions enable the child to develop second-order representations of their own subjective experiences, which positively impacts affect regulation and self-control. The capacity to reflect on mental states provides the foundation for these regulative processes [17]. From this perspective, mentalizing abilities and attachment relationships are seen as loosely interconnected [18,19]. Moreover, empirical evidence strongly supports the idea that both the attachment and mentalizing processes play central roles in stress and emotion regulation [20].

## Mentalizing capacity and development

Mentalizing is a critical skill that helps people navigate the complex social world they belong to throughout their lives. Cognitive abilities acquired during adolescence contribute to a more nuanced understanding of how changes in mental states are expressed. Beyond just attributing emotional states to oneself and others, adolescents can develop more intricate interpretations of mental states, emotions, and behaviors [21]. However, these new skills can be fragile, with a tendency to revert under stress or when encountering unfamiliar and intense emotional states, as the world becomes increasingly complicated and confusing [8,22]. Fonagy et al. [23] refer to a reaction against mentalizing in adolescence, mentioning how adolescents may withdraw from interactions or mentalizing altogether, leading to a more impulsive, insensitive mindset.

Research suggests that mentalizing tends to improve with age, with evidence showing changes throughout adolescence and into early adulthood. Particularly, mentalizing ability increases gradually from early adolescence to young adulthood and keeps mature beyond it, aligning with brain development. Notably, females aged 17–18 showed different mentalizing patterns compared to those over 20 [8]. In this line of thinking, Poznyak et al., [24] demonstrate that while basic mentalizing processes exist in adolescence, as people get older, they become more flexible in how they shift between the "self-other" representations.

Despite growing interest in the relationship between attachment and student learning and adjustment, research on the way mentalizing operates remains limited. Only one study has emphasized the significance of mentalizing for students' academic success and well-being in a counseling setting [25]. This gap in the literature underscores the need for further research into how mentalizing affects student outcomes and adjustment in educational contexts.

## Mentalizing and related concepts

Mentalizing covers various aspects of social cognition, overlapping with psychological constructs like empathy, emotional intelligence, mind-blindness, and psychological mindedness (PM) [20]. PM is described as "a person's ability to see relationships among thoughts, feelings, and actions with the goal of understanding the meaning and causes of their experiences and behavior" [26]. It reflects both the interest in and the capacity to think about affect, thoughts, and behaviors in a unified manner, while it aligns with mentalizing in its emotional and cognitive aspects, such as cognitive styles [27], used to understand others' behavior.

Moreover, PM, like mentalizing, is linked to attachment; individuals with secure attachment [28,29] and those who are well-adjusted are more likely to show higher levels of PM. It has also been associated with peer attachment in university settings [27]. The key difference between PM and mentalizing lies in the implicit-explicit dimension. PM focuses more on explicit and conscious understanding of mental states, emphasizing the self and one's own mental state, while mentalizing involves implicit processes.

## Epistemic trust

Within the mentalizing system, Epistemic Trust (ET) has emerged as a construct that both facilitates and is facilitated by mentalizing [20]. ET refers to the ability to assess information from the social environment as accurate, reliable, and

relevant [30]. This capacity allows individuals to integrate new information into their existing knowledge base [31]. Mirrored emotional experiences tend to enhance epistemic trust, establishing a link between ET and attachment. In contrast, a pattern of unmirrored emotional experiences can create epistemic hypervigilance, potentially leading to epistemic mistrust [31,20]. Attachment avoidance may lead to epistemic mistrust, while anxious attachment may foster epistemic uncertainty due to an overreliance on the perspective of attachment figure [30].

Mikulincer & Arad [32] proposed that individuals with insecure attachment are more easily threatened by information that challenges their existing beliefs because their sense of self is vulnerable—particularly to emotional overwhelm— which prompts them to seek knowledge stability to reduce arousal [30].

ET has also been conceptualized as an evolutionary capacity for trusting others as sources of social information, fostering resilience through social learning and deriving value from a steady flow of relevant information within the social environment. Secure relationships with caregivers during childhood support the development of ET, allowing it to be generalized to new relationships and contexts. Throughout life, peers, community members, and broader sociocultural influences can either promote or hinder the development of ET [20].

Recent studies have highlighted the protective role of mentalizing and epistemic trust against stress and emotional dysregulation in adolescence. Additionally, adaptive ER was predicted by self-focused mentalizing [33]. Although research has indicated that the attachment system regulates emotions to maintain security or manage insecurity, recent studies suggest bidirectional influences between attachment and emotion regulation [34]. However, the relationship between emotion regulation and ET remains underexplored.

## Emotion regulation

Emotion regulation (ER) refers to the conscious or unconscious efforts people make to influence which emotions they have, when they have them, and how they express and experience them [35]. A significant portion of research has focused on cognitive reappraisal and suppression. Reappraisal involves reframing a situation in a way that alters one's emotional response, while suppression entails inhibiting the outward expression of emotion [36,37]. Reappraisal is often seen as adaptive, and suppression is viewed as maladaptive. However, recent literature indicates that the context in which the emotion emerges is crucial [38,39]. Additionally, suppression has been linked to positive outcomes [40]. Habitual use of reappraisal correlates with greater positive emotion and less negative emotion, while frequent suppression leads to less positive emotion and more negative emotion. Reappraisal is also associated with better interpersonal functioning and well-being, whereas suppression is typically related to poorer interpersonal functioning and lower well-being [36].

## Emotion regulation and development

: Studies suggest that ER strategies evolve throughout adulthood [41], with significant changes during periods of transition, such as emerging adulthood [42]. As individuals age, reappraisal tends to increase while suppression decreases, reflecting a healthier pattern of emotion regulation [36]. By the end of adolescence and into emerging adulthood, cognitive functions become more sophisticated, leading to more flexible ER. This flexibility results in greater coherence in recognizing and understanding one's own and others' emotions, a better understanding of the selectivity of personal perceptions and evaluations, and more insight into one's emotional behavior [43]. These changes are part of the broader emotional development process; studies show that the cognitive control aspects of reappraisal become more active with age [44].

Reappraisal involves both cognitive control processes and representing the mental states of oneself and others, especially when attending to one's own emotional state or reconsidering others' during the reappraisal process [45,46]. This perspective suggests that mentalizing and ER might mediate the relationship between attachment and epistemic trust. Numerous studies have linked ER to attachment in various contexts, including education, especially higher education

[47,48]. Both attachment and ER are considered interrelated constructs that can help explain a variety of factors that impact students' academic life [49,50].

## Emotion regulation and psychological constructs in higher education

The frequent use of reappraisal has been associated with behaviors that promote academic achievement [51]. Recent research underscores the role of ER in students' approaches to learning [52,53]. Surface learners tend to score low on reappraisal and high on suppression, while deep learners exhibit the opposite pattern [4]. ER also serves as a precursor to academic emotions, allowing students to effectively manage their emotions during learning [35,54].

Several studies have examined learning profiles that encompass emotional and cognitive factors like cognitive ER, implicit ER [2,4], well-being, psychological strengths, and mental health variables [3,5]. These studies suggest that sub-optimal and dissonant profiles are common in the first two years of study. Other research has focused on learning profiles that use attachment orientation alongside elements of emotion regulation, academic emotions, and learning, but these studies did not identify significant differences across years of study, potentially indicating the central role of attachment in shaping psychological constructs [55].

Research on "relaxed" dissonant profiles, where students do not experience negative emotional tension despite poor achievement, suggests that these profiles might not be harmful. They may represent a developmental phase in students' academic journey, during which they develop new, more functional processes [56]. From our perspective, these profiles could signify a regression to a less reflective state [57], with students withdrawing from interactions. This observation points to the crucial role of mental capacity in student learning.

Additionally, the recent focus on the interplay between learning and psychological strengths, like ER and Sense of Coherence [58], aligns with the conceptualization of the mentalizing processes. In the context of these processes, trust in others as sources of social information fosters resilience through a salutogenic process, allowing individuals to gain from the stream of relevant information in the social environment [20].

Our perspective that exploring the attachment and mentalizing processes can inform understandings of learning, success, and adjustment in higher education aligns with Moreau et al.'s [13] caution against "overstating the environmental factors in success" compared to individual traits. Recognizing factors that are difficult to change allows for resource allocation where they can make a difference, accommodating individual needs to enable meaningful interventions that help students thrive (see also 4). Understanding the mediating role of psychological strengths and difficulties in different years of study highlights the need for interventions targeted at specific student groups.

## Hypotheses and focus of the study

This study examines the distinctions between junior and senior university students in terms of affect regulation, including cognitive emotion regulation strategies like reappraisal and suppression, psychological mindedness, mentalizing, and attachment. These factors are known to correlate with student adjustment and mental health. Given the enhanced cognitive capacities in young adults, we anticipate that the relationship between insecure attachment orientation and ET is mediated through most of constructs that differentiate junior from senior students.

Based on the Desatnik et al. study [8], it is hypothesized that senior students will show higher levels of emotion regulation, psychological mindedness and mentalizing compared to junior students (H1). Furthermore, junior students are expected to report that the negative aspects of mentalizing (uncertainty/confusion) mediate the relationship between insecure attachment and ET (H2), and senior students will report that the positive aspects of mentalizing (certainty, interest/curiosity) mediate the relationship between insecure attachment and ET (H3). Suppression is expected to mediate the relationship between avoidance attachment and ET for both junior and senior students (H4); reappraisal is hypothesized to mediate the relationship between anxious attachment and ET only for senior students (H5). Suppression will be a dominant mediator in the relationship between avoidant attachment and epistemic trust (H6).

## Method

### Participants and procedure

The present study is part of a larger research project which used community and university sample, and was conducted between October 3rd, 2020, and April 30th, 2021, through social media (Facebook, Viber, MS Messenger) or university advertisements. Participants met inclusion criteria based on age (over 17 years) and language (fluent Greek speakers). They all completed the same set of questionnaires after providing written informed consent.

In the present study the university sample was used, consisting of 460 undergraduate students. Most of the participants were Greek nationals (n = 446; 97%), with most attending the University of Ioannina (n = 383; 83%), especially the Social Sciences and Education departments (Psychology, Preschool Studies etc). That said, participants were primarily female (n = 440; 96%), which is common in social and education studies. Age ranged from 17 to 23 years (mean age 20.2 years, SD = 1.14). Of these, 216 (47%) were first- or second-year students (juniors), while 244 (53%) were in their final two years (seniors).

This study received ethical approval from the University of Ioannina Research Ethics Committee (35299/30-09-2020) and was conducted in accordance with ethical standards and guidelines, ensuring the protection of participants' rights, privacy, and well-being. All data collected was kept confidential and used solely for the purposes of this research.

### Measures

The Greek version of the Reflective Functioning Questionnaire (short form) (RFQ) [59] was used to measure mentalizing. This 31-item scale evaluates mentalizing abilities by asking respondents about mental processes related to themselves and others. It encompasses three subscales representing different levels of mentalizing: certainty, interest/curiosity, and uncertainty/confusion. An example item for certainty is "It's easy for me to figure out what someone else is thinking or feeling." For interest/curiosity, an example item is "I pay attention to the impact of my actions on others' feelings," while uncertainty/confusion includes items like "Strong feelings often cloud my thinking." Participants responded on a 7-point Likert scale ranging from "strongly disagree" (1) to "strongly agree" (7).

The Greek adaptation of the Epistemic Trust, Mistrust, and Credulity Questionnaire (ETMCQ) was used to study epistemic trust [60,59].The ETMCQ consists of 15 items across three subscales: epistemic trust (e.g., "I usually ask people for advice when I have a personal problem"), epistemic credulity (e.g., "I am often considered naïve because I believe almost anything that people tell me"), and epistemic mistrust (e.g., "I often feel that people do not understand what I want and need"), with 5 items each. Responses are on a 7-point Likert scale from "strongly disagree" (=1) to "strongly agree" (=7).

The Emotion Regulation Questionnaire (ERQ) [61] was used to measure cognitive emotion regulation, in its Greek adaptation [59]. It is a 10-item measure that assesses two common emotion regulation strategies: cognitive reappraisal (6 items) and expressive suppression (4 items). Responses are on a 7-point Likert scale, ranging from "strongly disagree" (=1) to "strongly agree" (=7). Example items for cognitive reappraisal include statements like "When I want to feel more positive emotion (such as joy or amusement), I change what I'm thinking about," while for expressive suppression, an example is "I keep my emotions to myself."

The Experiences in Close Relationships-Revised (ECR-RD12) [62] is a 12-item measure assessing individual differences in two attachment dimensions which measure insecure attachment: (a) attachment-related anxiety and (b) attachment-related avoidance. Each subscale consists of 6 items rated on a 7-point Likert scale, ranging from "strongly disagree" (=1) to "strongly agree" (=7). An example item for attachment-related anxiety is, "I'm afraid that once a romantic partner gets to know me, he or she won't like who I really am." For attachment-related avoidance, an example item is, "I prefer not to be too close to romantic partners." In the present study the Greek adaptation of the questionnaire was used [59].

The Psychological Mindedness scale is a 45-item instrument that assesses an individual's capacity to identify relationships among feelings, actions, and thoughts, aiming to understand the meanings and causes of their experiences and

behavior [63]. Example items include, "I would be willing to talk about my personal problems if I thought it might help me or a member of my family" and "Often I don't know what I'm feeling." Items are rated on a 4-point scale, from "strongly agree" (=1) to "strongly disagree" (=4). The scale was used in its unidimensional form, with higher scores on the scale indicate greater psychological mindedness.

## Statistical analyses

Correlational analyses of ETMCQ with developmental and psychological measures were examined. The means, standard deviations, and the correlation matrix between the variables in the study were calculated first for the total sample and then for the two distinct groups of students (junior/senior). Values up to 0.19 were considered "slight", from 0.20 to 0.39 "low", from 0.40 to 0.69 "moderate" and over 0.70 "high" [64,65]. The internal consistency of the scales (reliability) was tested with Cronbach's *alpha* coefficient [66]. Missing data were handled using listwise deletion.

Mediation hypotheses were tested using the PROCESS Procedure for SPSS (v. 4.21), a conditional process modeling program that applies an ordinary least-squares or logistic-based path analytical framework to evaluate both direct and indirect effects with observed composite scores as proxies for latent variables [67]. All outcome variables were continuous, and the standard assumptions of ordinary least squares regression (normality, independence, and homoscedasticity) were tested [68]. Bias-corrected bootstrap tests with a 95% confidence interval were conducted to determine the significance of indirect effects. Using random samples with replacement from the original data set, 5,000 bootstrap samples were generated. This method is used for the control of the results' stability. To interpret the mediation models, the Zhao et al. approach was used (i.e., identification of three patterns consistent with mediation and two with non-mediation) [69]. All inferential analyses used the Statistical Package for Social Science (SPSS v.29) with a critical p-value of <.05.

## Results

### Reliability and validity of the scales

Internal consistency indices were found satisfactory for most of the scales and subscales, except for the Interest/curiosity, $α = 0.60$, and the Mistrust subscale $α = 0.53$, approximating the indices of the original scales (Table 1). Since the data were taken from a greater study, convergent and structural validity of the Greek versions of the original scales have been discussed elsewhere and have been found to have satisfactory psychometric properties [59].

### Descriptive and correlational analyses

The percentiles for students' responses on the Experiences in Close Relationships-Revised (ECR-R) scale were calculated to ensure that the sample met community criteria as to the attachment style (Table 2). Percentages were found to be distributed evenly. Nearly 60% (n = 268) scored above the 50th percentile in the anxious attachment dimension, while 52% (n = 237) scored above the 50th percentile in the avoidant attachment dimension. In total, 35% (n = 162) scored above the 50th percentile on both attachment dimensions.

For the Anxious dimension the cut-off values are 25% = 2.80; 50% = 3.60; 75% = 4.80. For the Avoidant dimension the cut-off values are 25% = 1.86; 50% = 2.57; 75% = 3.29.

Attachment orientations were correlated with most mentalizing dimensions and epistemic trust factors in the total sample (Table 3). Specifically, anxious attachment orientation showed significant low correlations with reappraisal, $r = −0.21$, $p < 0.001$, and suppression, $r = 0.25$, $p < 0.001$. Concerning epistemic trust, anxious attachment orientation correlated moderately with mistrust, $r = 0.49$, $p < 0.001$, and credulity, $r = 0.40$, $p < 0.001$; however, the correlation with trust was non-significant, $r = −0.01$, $p = 0.754$. Uncertainty/confusion was moderately and significantly correlated with anxious attachment orientation, $r = 0.44$, $p < 0.001$. The correlation with psychological mindedness was found low and significant, $r = −0.30$, $p < 0.001$. Avoidant attachment style was moderately correlated with suppression, $r = 0.43$, $p < 0.001$, and only slightly

**Table 1. Cronbach *alpha*s for the total scales and subscales of the study.**

| Scale & subscales | Cronbach *α* | Cronbach *α* (original scale) |
|---|---|---|
| ECR-RD12 (total scale) | 0.80 | – |
| *Anxiety-related attachment* | 0.81 | 0.88 |
| *Avoidance-related attachment* | 0.81 | 0.87 |
| ERQ (total scale) | 0.73 | – |
| *Reappraisal* | 0.86 | 0.75 to 0.82 |
| *Suppression* | 0.76 | 0.68 to 0.76 |
| RFQ (total scale) | 0.79 | 0.79 |
| *Excessive certainty* | 0.85 | 0.88 |
| *Uncertainty/confusion* | 0.87 | 0.87 |
| *Interest/curiosity* | 0.60 | 0.68 |
| ETMCQ (total scale) | 0.74 | 0.71 to 0.78 |
| *Trust* | 0.75 | 0.69 to 0.81 |
| *Mistrust* | 0.53 | 0.65 to 0.72 |
| *Credulity* | 0.73 | 0.75 to 0.81 |
| Psychological Mindedness (total scale) | 0.84 | 0.80 |

**Table 2. Distribution of the students' answers in the ECR-R.**

| | Anxious | | Avoidant | |
|---|---|---|---|---|
| Percentiles | *n* | % | *n* | % |
| <25% | 98 | 21,3 | 107 | 23,3 |
| 25-50% | 94 | 20,4 | 116 | 25,2 |
| 50-75% | 142 | 30,9 | 109 | 23,7 |
| >75% | 126 | 27,4 | 128 | 27,8 |

with reappraisal, $r=-0.17$, $p<0.001$. Negative low correlations were found with trust, $r=-0.36$, $p<0.001$, with interest/curiosity, $r=-0.23$, $p<0.001$, and psychological mindedness, $r=-0.46$, $p<0.001$. Reappraisal correlated positively with trust, $r=0.22$, $p<0.001$, interest/curiosity, $r=0.34$, $p<0.001$, and psychological mindedness, $r=0.30$, $p<0.001$. Low correlations were found between suppression and trust, $r=-0.35$, $p<0.001$, mistrust, $r=0.33$, $p<0.001$, credulity, $r=0.21$, $p<0.001$, uncertainty/confusion, $r=0.20$, $p<0.001$, and psychological mindedness, $r=-0.46$, $p<0.001$. Trust correlated slightly with interest/curiosity, $r=0.38$, $p<0.001$, and moderately with psychological mindedness, $r=0.56$, $p<0.001$. Mistrust correlated moderately with uncertainty/confusion, $r=0.46$, $p<0.001$, and lowly with psychological mindedness, $r=-0.30$, $p<0.001$. Credulity was moderately correlated with uncertainty/confusion, $r=0.47$, $p<0.001$.

To identify possible deviations, correlations were also examined in separate samples for junior and senior students. For emotion regulation, avoidant attachment orientation was significantly correlated with reappraisal only for senior students $r=-0.21$, $p<0.001$. Regarding ET, there were slight but significant correlations between mistrust and reappraisal, $r=-0.15$, $p=0.017$, and between credulity and psychological mindedness, $r=-0.29$, $p<0.001$ only for senior students. When examining mentalizing dimensions, significant correlations were observed for junior students between excessive certainty and mistrust, $r=0.15$, $p=0.032$, interest/curiosity and mistrust, $r=0.14$, $p=0.045$, uncertainty/confusion and trust, $r=0.16$, $p=0.020$. For senior students significant associations were identified between excessive certainty and anxiety-related attachment, $r=-0.14$, $p=0.031$, avoidance-related attachment, $r=-0.21$, $p<.001$, reappraisal, $r=0.22$, $p<0.001$, and trust, $r=0.28$, $p<0.001$; uncertainty/confusion correlated positively with avoidance-related attachment, $r=0.22$, $p<0.001$,

**Table 3. Correlations of the variables and Means comparison between junior and senior undergraduate students.**

| Variables | 1 | 2 | 3 | 4 | 5 | 6 | 7 | 8 | 9 | 10 | 11 |
|---|---|---|---|---|---|---|---|---|---|---|---|
| *Attachment orientations* | | | | | | | | | | | |
| 1.Anxious | – | | | | | | | | | | |
| 2.Avoidant | | | | | | | | | | | |
| Total | 0.27** | – | | | | | | | | | |
| Junior | 0.19** | – | | | | | | | | | |
| Seniors | 0.33*** | – | | | | | | | | | |
| *Emotional regulation* | | | | | | | | | | | |
| 3.Reappraisal | | | | | | | | | | | |
| Total | −0.21*** | −0.17*** | – | | | | | | | | |
| Junior | −0.16* | −0.11 | – | | | | | | | | |
| Seniors | −0.26*** | −0.21*** | – | | | | | | | | |
| 4.Suppression | | | | | . | | | | | | |
| Total | 0.25*** | 0.43*** | −0.05 | – | | | | | | | |
| Junior | 0.27*** | 0.42*** | −0.06 | – | | | | | | | |
| Seniors | 0.22*** | 0.44*** | −0.14* | – | | | | | | | |
| *Epistemic trust* | | | | | | | | | | | |
| 5.Trust | | | | | | | | | | | |
| Total | −0.03 | −0.36*** | 0.22*** | −0.35*** | – | | | | | | |
| Junior | −0.02 | −0.35*** | 0.20** | −0.27*** | – | | | | | | |
| Seniors | −0.07 | −0.36*** | 0.23** | −0.42*** | – | | | | | | |
| 6.Mistrust | | | | | | | | | | | |
| Total | 0.49*** | 0.16* | −0.10* | 0.33*** | −0.01 | – | | | | | |
| Junior | 0.46*** | 0.17* | −0.06 | 0.34*** | 0.10 | – | | | | | |
| Seniors | 0.52*** | 0.16* | −0.15* | 0.32*** | 0.12 | – | | | | | |
| 7.Credulity | | | | | | | | | | | |
| Total | 0.40*** | 0.10* | −0.07 | 0.21*** | 0.09 | 0.43*** | – | | | | |
| Junior | 0.26*** | 0.06 | −0.03 | 0.23*** | 0.14* | 0.35*** | – | | | | |
| Seniors | 0.50*** | 0.13* | −0.12 | 0.20*** | 0.03 | 0.47*** | – | | | | |
| *Mentalizing* | | | | | | | | | | | |
| 8.Excessive certainty | | | | | | | | | | | |
| Total | −0.10* | −0.13* | 0.17** | 0.03 | 0.13** | 0.08 | −0.13** | – | | | |
| Junior | −0.06 | −0.04 | 0.10 | 0.09 | −0.02 | 0.15* | −0.13 | – | | | |
| Seniors | −0.14* | −0.21*** | 0.22*** | −0.02 | 0.28*** | 0.02 | −0.14* | – | | | |
| 9.Interest/Curiosity | | | | | | | | | | | |
| Total | −0.03 | −0.22*** | 0.26*** | −0.18*** | 0.38*** | 0.08 | −0.01 | 0.28*** | – | | |
| Junior | 0.01 | −0.29*** | 0.24*** | −0.19*** | 0.39*** | 0.14* | −0.04 | 0.25*** | – | | |
| Seniors | −0.06 | −0.16* | 0.26*** | −0.17** | 0.36*** | 0.02 | 0.01 | 0.29*** | – | | |
| 10.Uncertainty/Confusion | | | | | | | | | | | |
| Total | 0.44*** | 0.13** | −0.16*** | 0.20*** | 0.05 | 0.46*** | 0.47*** | −0.10* | 0.03 | – | |
| Junior | 0.42*** | 0.04 | −0.09 | 0.19** | 0.16* | 0.47*** | 0.38*** | −0.07 | 0.06 | – | |
| Seniors | 0.46*** | 0.22*** | −0.23*** | 0.21*** | −0.06 | 0.45*** | 0.54*** | −0.14* | −0.12 | – | |
| 11.*Psychological Mindedness* | | | | | | | | | | | |
| Total | −0.30*** | −0.46*** | 0.30*** | −0.46*** | 0.56*** | −0.30*** | −0.22*** | 0.22*** | 0.44*** | −0.33*** | – |
| Junior | −0.20*** | −0.47*** | 0.32*** | −0.37*** | 0.60*** | −0.15* | −0.05 | 0.18** | 0.53*** | −0.16* | |
| Seniors | −0.35*** | −0.53*** | 0.38*** | −0.50*** | 0.61*** | −0.33*** | −0.29*** | 0.31*** | 0.47*** | −0.35*** | |

*(Continued)*

**Table 3.** (Continued)

| Variables | 1 | 2 | 3 | 4 | 5 | 6 | 7 | 8 | 9 | 10 | 11 |
|---|---|---|---|---|---|---|---|---|---|---|---|
| Mean | | | | | | | | | | | |
| Total | 3.82 | 2.59 | 4.82 | 3.21 | 5.48 | 4.36 | 3.60 | 45.31 | 33.92 | 54.04 | 3.08 |
| Junior | 3.87 | 2.63 | 4.71 | 3.27 | 5.39 | 4.33 | 3.53 | 44.43 | 33.53 | 53.86 | 3.07 |
| Seniors | 3.79 | 2.55 | 4.92 | 3.16 | 5.56 | 4.39 | 3.66 | 46.14 | 34.23 | 54.12 | 3.09 |
| Standard deviation | | | | | | | | | | | |
| Total | 1.36 | 0.98 | 1.13 | 1.20 | 0.91 | 0.95 | 1.23 | 8.91 | 4.36 | 14.92 | 0.28 |
| Junior | 1.37 | 1.00 | 1.17 | 1.16 | 0.97 | 0.88 | 1.07 | 9.34 | 4.53 | 14.42 | 0.28 |
| Seniors | 1.36 | 0.95 | 1.08 | 1.23 | 0.84 | 1.01 | 1.36 | 8.44 | 4.18 | 15.38 | 0.27 |
| *t*-test (*df* = 458) | 0.632 | 0.888 | −2.013* | 1.000 | −1.945* | −0.647 | −1.089 | −2.071* | −1.727* | −0.188 | −0.569 |
| Cohen's *d* | 0.06 | 0.08 | −0.19 | 0.09 | −0.18 | −0.06 | −0.10 | −0.19 | −0.16 | −0.02 | −0.05 |

Note: *p < .05; ** p < .01; *** p < .001.

and negatively with reappraisal, $r = -0.23$, $p < 0.001$. Psychological mindedness showed similar correlations for both age groups.

These results suggest that the two groups may differ in the research variables. Thus, mean comparisons were conducted to investigate this further. Mean differences between junior (first and second year) and senior (third and fourth year) students are reported in Table 3. Significant differences were identified in the "positive" dimensions of mentalizing, epistemic trust, and emotion regulation. Senior students reported higher levels of certainty, $t(458) = -2.071$, $p = 0.019$; interest/curiosity, $t(458) = -1.727$, $p = 0.042$; epistemic trust, $t(458) = -1.945$, $p = 0.026$; and reappraisal, $t(458) = -2.013$, $p = 0.022$.

### Mediational analyses

The results of the correlational analyses and means comparison were indicative of a possible mediational relation between attachment, emotional regulation, mentalizing, and epistemic trust across the two groups of students. Psychological mindedness, on the other hand, was found similar in the two groups and was not included in the mediation model.

**Assumptions.** First, the necessary assumptions for regression analysis were investigated. Independence of observations was examined with the Durbin-Watson statistic test [70]. Values between 1.5 and 2.5 suggest that the residuals in the models are independent [71]. The Durbin-Watson statistics for the three dimensions—1.938, 2.004, and 1.925—indicate that the assumption of independent residuals is met. Linearity in the relationships among the variables was examined by plotting the independent and dependent variables using the studentized residuals and unstandardized predicted values. Fig 1 shows scatterplots for trust (1a), mistrust (1b), and credulity (1c). The data appear horizontal through visual inspection, suggesting that the relationships between variables are linear. Homoscedasticity of error values (i.e., the variability in the dependent variable not attributable to the independent variables is consistent across different scores of the dependent variable) is checked by the PROCESS macro, which uses the HC3 (Davidson-McKinnon) test, which is a reliable inference method without assuming equal variance of estimation errors [72,73]. Multicollinearity among the independent variables was examined using the variance inflation factor (VIF) and the tolerance statistic. VIF should not exceed 10, and tolerance should be above 0.10 [74]. In the three models, VIF values for the independent variables ranged between 1.012 and 1.364, and tolerance values ranged between 0.733 and 0.988, indicating an absence of multicollinearity. Finally, to that residuals are normally distributed, we visually inspect histograms with a superimposed normality curve and the probability-probability (P-P) plots in Fig 2. The distribution seems evenly skewed, with points aligned along the diagonal line, suggesting a normal distribution, indicating no need for further transformation.

**Fig 1. Scatter diagrams for the three dimensions of epistemic trust representing a linear relationship among all variables in the models.**

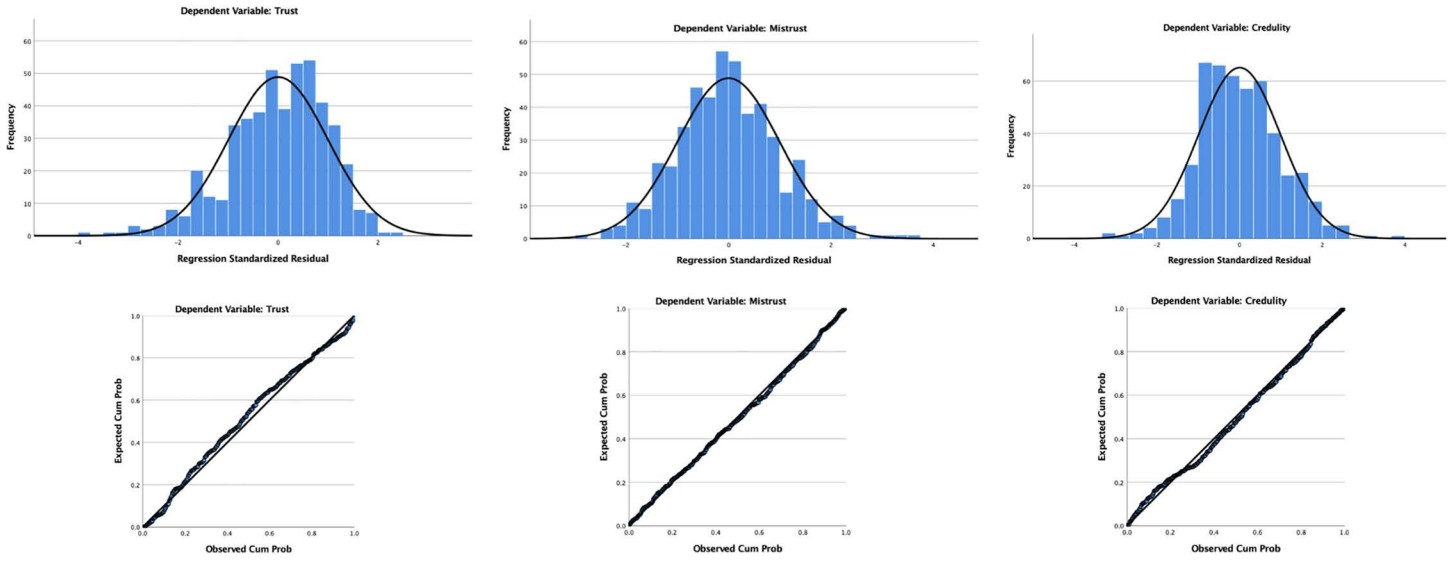

**Fig 2. Histograms with normality curve and P-P plots for the three dimensions of epistemic trust.**

**Conditional analyses.** Parallel mediations were conducted, assuming that the three mentalizing dimensions and the two emotion regulation dimensions mediate the relationship between attachment styles and epistemic trust constructs similarly. These models were tested separately for junior and senior students (Figs 3–6). Simple mediation tests indicated that mentalizing and emotion regulation dimensions had a statistically significant mediating effect on the relationship between attachment orientations and epistemic trust dimensions, differing for the two age groups. Including these factors in the model explained 26 percent of the variation for trust in juniors, $R^2 = 0.26$; 37 percent for mistrust, $R^2 = 0.37$; and 16 percent for credulity, $R^2 = 0.16$. In seniors, the factors explained 31 percent for trust, $R^2 = 0.31$; 36 percent for mistrust, $R^2 = 0.36$; and 37 percent for credulity, $R^2 = 0.37$.

For anxious attachment in junior students (Fig 3), uncertainty/confusion was a significant mediator for the direct effect on mistrust, $b = 0.14$, $SE = 0.03$, 95% CI: 0.08, 0.20, and on credulity, $b = 0.14$, $SE = 0.04$, 95% CI: 0.07, 0.21. Juniors with higher levels of anxious attachment reported higher uncertainty/confusion, which was associated with increased mistrust and credulity. A no-effect pattern, where neither direct nor indirect effect existed, was observed between anxious attachment and trust, mediated by uncertainty/confusion, $b = 0.08$, $SE = 0.03$, 95% CI: 0.02, 0.14. Higher levels of uncertainty/

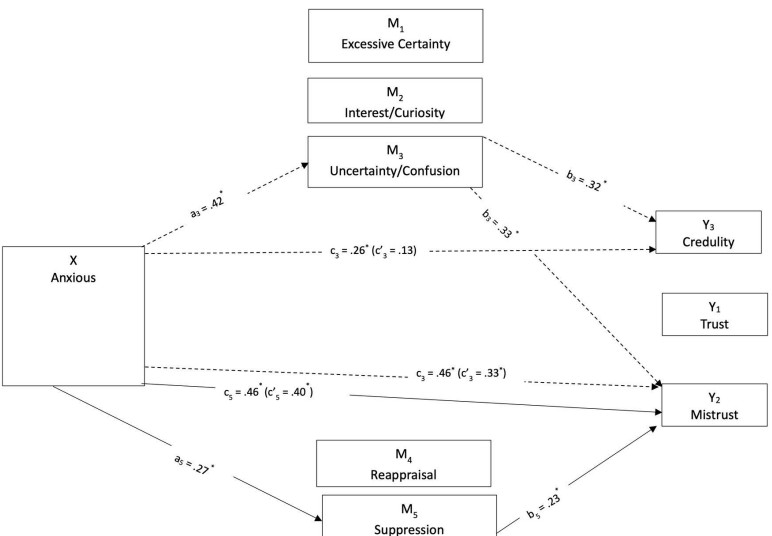

**Fig 3. Direct and indirect effects of anxious attachment style on epistemic trust dimensions mediated by mentalizing and emotional regulation for junior students.**

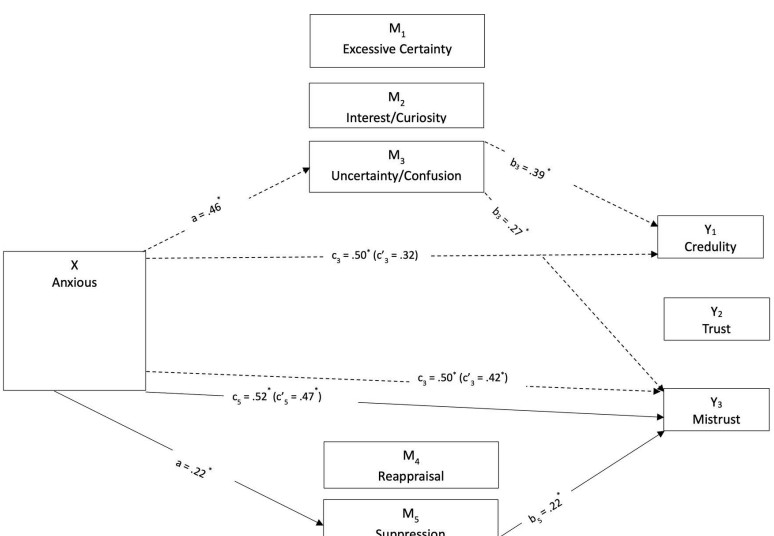

**Fig 4. Direct and indirect effects of anxious attachment style on epistemic trust dimensions mediated by mentalizing and emotional regulation for senior students.**

confusion mediated the non-significant effect of anxious attachment on trust, shifting the direction from slightly negative to slightly positive.

The direct effect on mistrust increased via suppression, $b = 0.06$, $SE = 0.02$, 95% CI: 0.02, 0.11. Higher levels of anxious attachment were associated with greater suppression, which was linked to increased mistrust. There was also a no-effect pattern between anxious attachment and trust, mediated by suppression, $b = -0.08$, $SE = 0.03$, 95% CI: −0.15, −0.03. This could be attributed to the opposing associations of the a path (positive) and the b path (negative) with trust. Higher levels of suppression mediated the insignificant effect of anxious attachment on trust, slightly reducing the positive direct effect.

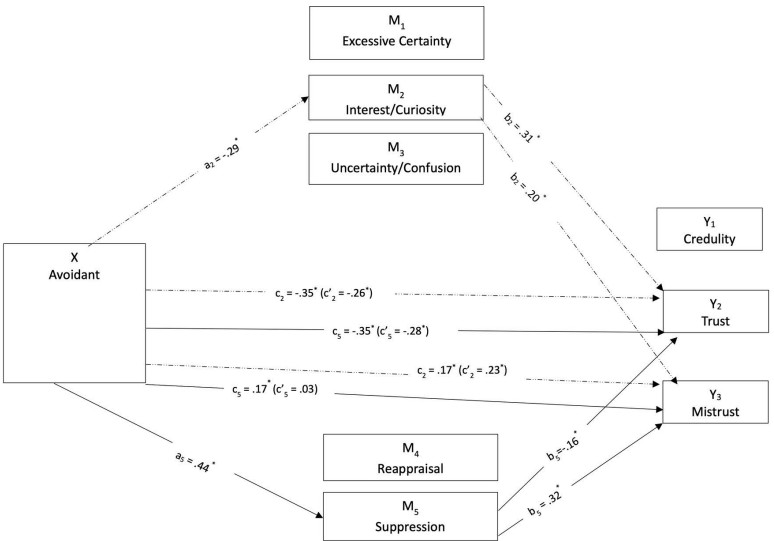

**Fig 5. Direct and indirect effects of avoidant attachment style on epistemic trust dimensions mediated by mentalizing and emotional regulation for junior students.**

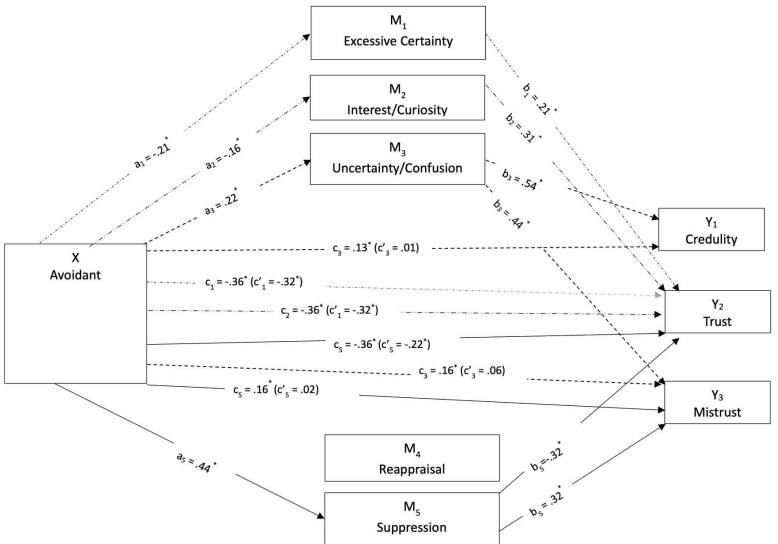

**Fig 6. Direct and indirect effects of avoidant attachment style on epistemic trust dimensions mediated by mentalizing and emotional regulation for senior students.**

Similarly, a no-effect pattern was identified between anxious attachment and trust, mediated by reappraisal, $b = -0.03$, $SE = 0.02$, 95% CI: −0.08, −0.01. This could also be explained by the opposing associations of the a path (negative) and the b path (positive) with epistemic trust. Higher levels of reappraisal mediated the non-significant effect of anxious attachment on epistemic trust, slightly decreasing the positive direct effect.

A similar pattern emerged for senior students (Fig 4). Uncertainty/confusion mediated the direct effect on mistrust, $b = 0.12$, $SE = 0.03$, 95% CI: 0.07, 0.19, and credulity, $b = 0.18$, $SE = 0.03$, 95% CI: 0.12, 0.24. Senior students' higher levels

of anxious attachment correlated with increased mistrust and credulity when mediated by uncertainty/confusion. However, there was a no-effect pattern between anxious attachment and epistemic trust, which was nonetheless mediated by certainty, $b = -0.04$, $SE = 0.02$, 95% CI: −0.08, −0.01. Higher levels of certainty changed the relationship between anxious attachment and epistemic trust from slightly negative to slightly positive. Further, suppression mediated the direct effect of anxious attachment on mistrust, $b = 0.05$, $SE = 0.02$, 95% CI: 0.02, 0.09. Students with higher levels of anxious attachment reported greater suppression, which was linked to increased mistrust. There was also a no-effect pattern between anxious attachment and epistemic trust, mediated by suppression, $b = -0.09$, $SE = 0.03$, 95% CI: −0.16, −0.04.

The mediation of the relationship between avoidant style and epistemic trust by mentalizing and emotion regulation was tested similarly. For junior students' avoidant style (Fig 5), interest/curiosity mediated the direct effect on epistemic trust $b = -0.09$, $SE = 0.03$, 95% CI: −0.16, −0.04, and mistrust, $b = -0.06$, $SE = 0.03$, 95% CI: −0.12, −0.02. Juniors' lower levels of avoidant attachment related to higher interest/curiosity increased the negative effect of avoidant attachment on epistemic trust and decreased the positive effect on mistrust. Suppression also mediated the direct effect of avoidant attachment on epistemic trust and mistrust, $b = -0.07$, $SE = 0.03$, 95% CI: −0.14, −0.01, and $b = 0.14$, $SE = 0.04$, 95% CI: 0.07, 0.22, respectively. Higher levels of suppression among junior students reduced the negative effect of avoidant attachment on epistemic trust and minimized the positive effect on mistrust. A similar no-effect pattern was observed for the indirect effect on credulity, mediated by suppression, $b = 0.11$, $SE = 0.03$, 95% CI: 0.05, 0.18.

For senior students, the mediation patterns differed (Fig 6). First, the effect of avoidant attachment on epistemic trust was mediated by certainty, $b = -0.04$, $SE = 0.02$, 95% CI: −0.08, −0.01, and interest/curiosity, $b = -0.05$, $SE = 0.02$, 95% CI: −0.09, −0.01. Lower levels of avoidant attachment with higher levels of epistemic trust among seniors were mediated by greater certainty and interest/curiosity. Uncertainty/confusion mediated the indirect effect of avoidant attachment on mistrust $b = 0.10$, $SE = 0.03$, 95% CI: 0.04, 0.16, and credulity, $b = 0.12$, $SE = 0.04$, 95% CI: 0.05, 0.20. Senior students' higher levels of avoidant attachment correlated with increased mistrust and credulity when mediated by uncertainty/confusion. Finally, suppression provided complementary mediation on the relation between avoidant attachment and epistemic trust, $b = -0.14$, $SE = 0.03$, 95% CI: −0.21, −0.08, and indirect mediation on the avoidant-mistrust relation, $b = 0.14$, $SE = 0.04$, 95% CI: 0.07, 0.22. Higher levels of suppression among seniors increased the negative effect of avoidant attachment on epistemic trust and augmented the positive effect on mistrust.

## Discussion

This study examined the differences between junior and senior students in psychological constructs related to the effect of attachment orientation on ET, such as emotion regulation, psychological mindedness, and mentalizing. Differences between the two age groups were investigated and the mediating role of the specific psychological constructs was explored. The results showed that senior students scored higher on reappraisal, certainty, and interest/curiosity compared to their junior counterparts. Mediation patterns were generally similar for junior and senior students. Regarding anxious attachment, none of these constructs mediated its relationship with epistemic trust. However, for avoidant attachment, only certainty and interest/curiosity mediated the relationship with epistemic trust. Notably, reappraisal failed to mediate any of these relationships. Instead, uncertainty/confusion and suppression—which are considered the "negative" aspects of emotion regulation and mentalizing—played a key mediating role in the relationship between insecure attachment and epistemic trust, despite not differentiating junior and senior students.

The results of the present study confirm prior research showing that the capacity to understand mental states increases with age, indicating that the mentalizing system does not reach its peak until early adulthood [8]. Reappraisal tends to improve with age, especially during transitions like emerging adulthood [75]. The results support our hypothesis that senior students score higher on constructs related to understanding one's own and others' mental states compared to junior students. Seniors scored higher on certainty, interest-curiosity, and reappraisal, which aligns with earlier studies indicating that senior students have more adaptive learning profiles, including higher scores on reappraisal, need for

cognition, and implicit emotion regulation [2,4]. Surprisingly, there was no difference in psychological mindedness, suggesting that the explicit and conscious understanding of mental states may require advanced cognitive skills, potentially acquired early by university students. This finding contradicts previous studies reporting significant differences between ages 17–18 and 20+ [8]. Additionally, suppression, another aspect of cognitive emotion regulation, did not differ between juniors and seniors. This could be due to suppression requiring fewer cognitive resources, making it more accessible across ages. This finding conflicts with previous studies suggesting that suppression decreases with age, with emerging adults likely to report less suppression and more reappraisal [60]. Finally, uncertainty/confusion did not differ between junior and senior students, potentially highlighting its importance in insecure attachment orientation.

Our hypothesis that positive aspects of mentalizing (certainty, interest/curiosity) will mediate the relationship between insecure attachment and epistemic trust reported only by senior students is not confirmed. Junior and senior students reported similar patterns. For junior students, interest/curiosity reduced the positive association between attachment avoidance and mistrust. This "beneficial" mediation may reflect developmental limitations that could be cautiously – given the self-report methodology of the present study – linked to research showing that brain regions responsible for understanding mental states remain immature until late adolescence [17]. Difficulties in recognizing and labeling facial expressions [76,82] could hinder the effective operation of the mentalizing system. This notion of developmental limitations is supported by the absence of this mediation for senior students and the role of interest/curiosity in enhancing the negative relationship between avoidant attachment and epistemic trust. Senior students displayed similar findings. The increase in their mentalizing capacity was evidenced by the mediating role of certainty and interest/curiosity in the relationship between avoidant attachment orientation and epistemic trust. As with junior students, interest/curiosity reduced the negative relationship between avoidant orientation and epistemic trust. This similarity is further supported by the identical beta coefficient for the association between interest/curiosity and epistemic trust. Interest in others may trigger a defensive response, leading to distance from them, as it may not always be safe to imagine what someone else is thinking [17].

Moreover, excessive certainty enhanced the negative relationship between avoidant attachment and epistemic trust reported by seniors, possibly suggesting that the development of more complex "self-other" processes might raise stress levels and activate negative schemata regarding others, thus triggering defensive independence. Therefore, cognitive maturation possibly inhibits controlled mentalizing [17,22]. This is supported by the mediation role of confusion/uncertainty, which enhanced the positive relationship between avoidant attachment and mistrust and credulity. This mediation pattern is similar to that of anxious students, supporting partly our H3 hypothesis (regarding senior students). It is possible that avoidant defense strategies may break down when the attachment system is fully activated due to academic pressure, critical decisions about the future, and exploring potential career paths, which heavily tax cognitive and coping resources [77]. Under such stress and uncertainty, avoidant individuals may increasingly lose their self-reliance and defensive independence, seeking validation and acceptance from others, similarly to their anxious counterparts [78].

The study reinforces the challenges anxiously attached individuals face in regulating affect and benefiting from positive social environments. The findings reject our H2 hypothesis: neither certainty nor interest/curiosity mediate the relationship between anxious attachment orientation and ET reported by senior students. This suggests that the negative relationship between insecure attachment and ET remains dominant, limiting one's ability to connect and engage in safe learning with others. The only factor that mediated the relationship between anxious attachment orientation and mistrust and credulity was uncertainty/confusion, which did not differ between junior and senior students supporting H3 hypothesis. Uncertainty/confusion increased the positive relationship between anxious attachment and mistrust and credulity, observed in both junior and senior students.

The mediation by a construct that does not increase with age may indicate the preexisting issues associated with anxious attachment, such as a low threshold for rapid and intense activation of the attachment orientation and

corresponding deactivation of controlled mentalizing processes, leading to automatic mentalizing [17]. This explanation is further supported by the considerable mediation of confusion in the relationship between anxious attachment and credulity reported by seniors. Credulity is a dominant aspect of mentalizing in anxiously attached individuals, often leading to clinging behavior regardless of age. This aligns with the attachment literature, which indicates that individuals with anxious attachment exhibit a rapid and extensive surge of negative emotions [79]. The particularly strong associations between uncertainty/confusion and mistrust and credulity reported by both junior and senior students reflect this pattern.

Moreover, the stronger associations reported by seniors are consistent with existing literature suggesting that individuals with a hyperactive attachment strategy (anxious) tend to switch to automatic mentalizing more quickly and take longer to return to controlled mentalizing [17]. This pattern illustrates the challenges anxiously attached individuals face when attempting to engage in social learning environments and highlights the need for targeted interventions to support them in these contexts.

The failure of reappraisal to mediate the relationship between insecure attachment and any dimension of ET, despite significant correlations with trust, mistrust, and anxious attachment orientation reported by senior students rejects H4 hypothesis. This finding may indicate that the impact of insecure attachment impairs the developmental capacities acquired in adolescence, especially when metacognitive abilities are required in stressful emotional contexts. Universities are often reported as highly demanding environments [80,81]. Anxiously attached individuals may struggle to feel safe and think creatively, making it difficult for them to use reappraisal to reduce emotional intensity. This lack of comfort and engagement can lead to ambivalence and indecision, preventing them from making confident decisions or taking a clear course of action [82].

Similarly, reappraisal did not mediate the relationship between avoidant attachment orientation and epistemic trust, even though senior students showed associations between trust, mistrust, and avoidant attachment. Its weak associations and inability to mediate the relationship between avoidant attachment orientation and epistemic trust might point to a facet of defensiveness. Reappraisal requires recognizing threats and errors, which avoidant individuals tend to deny [83]. Furthermore, the failure of reappraisal to mediate the relationship between insecure attachment and epistemic trust may be explained by the dominant role of suppression as a mediator in this relationship: the findings reject H5 (reappraisal is hypothesized to mediate the relationship between anxious attachment and ET only for senior students), but support H6 (suppression will be a dominant mediator in the relationship between avoidant attachment and epistemic trust). Although suppression did not differentiate junior and senior students, it emerged as a key mediator in the relationship between anxious and avoidant orientations and epistemic trust for both groups. This could indicate that both junior and senior students find suppression to be an adaptive strategy, as it requires fewer cognitive resources than reappraisal. Beyond self-report methodology, neuroimaging studies also suggest that people with high suppression scores find it challenging to recruit cognitive resources for reappraisal [84].

The significant role of suppression in the avoidant attachment orientation (that confirms H5) is indicated by (a) its mediating role in the relationships between avoidant orientation and trust and mistrust for both juniors and seniors, (b) the identical magnitude of the relationship between suppression and mistrust (as indicated by the beta coefficient) for both groups, and (c) the strong mediation magnitude, especially for senior students. Suppression strengthens the negative association between avoidant orientation and trust, while also reinforcing the positive association between avoidance and mistrust. This suggests that suppression is used to keep attachment-related needs deactivated to maintain self-reliance, indicating an inauthentic self that avoids closeness [83]. This inhibits mutuality and cooperation, making it difficult to engage in social relationships, which affects collaboration and cooperative learning [17,85].

The unexpected mediating role of suppression in anxious attachment orientation (that partly rejects H5) typically a key strategy for avoidant individuals, could be explained by the large number of participants who scored similarly high in both attachment orientations (over 50%). This interpretation is supported by the relatively weak magnitude of the mediation of suppression for anxious attachment compared to avoidant orientation.

## Implications

This study challenges the educational literature's focus on first-year students' vulnerability by revealing similar mediation patterns in both junior and senior students with insecure attachment, indicating defensive responses and automatic mentalizing. The findings support the notion that poor epistemic trust might arise not only from actual threats but also from internal "mentalizing holes," a legacy of adversity and vulnerability stemming from attachment issues [17]. This supports Fonagy's argument that attachment and mentalizing systems are only loosely related, emphasizing the need for higher education literature to explore student characteristics that are difficult to change [13] along with learning constructs [4]. These findings also suggest treating both junior and senior university students as a potentially at-risk group [86].

From this perspective, our recommendation for comprehensive psychological support for students throughout the university years should consider preventive interventions to reduce suppression and improve negative emotion tolerance, focusing on developing cognitive abilities to benefit from positive influences in the environment. The sensitivity to emotional triggers in social contexts and reduced capability to reflect [85], particularly among insecurely attached individuals, underscore the vulnerability to peer pressure [17]. This vulnerability can be seen – although tentatively in the context of the present study – in combination with the still-developing mentalizing brain areas, to suggest a high likelihood of automatic rather than controlled mentalizing, reinforcing the idea that university students could be considered at risk. Senior avoidant students appear as vulnerable as their anxious counterparts, showing high levels of credulity, potentially amplified by confusion about their own and others' mental states. This strong mediation pattern should be a warning for both avoidant and anxious individuals, given the epidemiologically high percentage for both groups (around 25% avoidant and 18–20% anxious) [74], implying a need for universities to foster environments that support psychological capacity and facilitate positive learning experiences. Interventions aimed at enhancing epistemic trust can focus on fostering safe curiosity and exploratory behavior, emphasizing the importance of a genuine interest in one's own and others' mental states. This approach could encourage agency, reflection, and relationships, offering a potential path for protective and preventive interventions [17]. Considering mentalizing as not just an individual but a social process, universities can create networks of relationships that facilitate thinking and appropriate action through mirroring, particularly in stressful situations, which in turn supports learning and affiliation [17,85]. Furthermore, findings suggest that interventions aimed at enhancing mentalization should be tailored developmentally, especially in early-to-mid adolescence [8]. Mentalizing can inform practical interventions in higher education by fostering university students' reflective abilities through structured group settings, where they could explore interpersonal dynamics and emotional experiences in real time. Moreover, students through narrative writing tasks with guided reflection could enhance the development of self-other awareness and metacognitive skills [24].

## Limitations and future study recommendations

This study provides a foundation for further exploration of the role of the mentalizing system and affect regulation in student learning and adjustment [87]. However, it has some limitations that should be acknowledged. Firstly, the reliance on self-report methodology, although commonly used, limits the understanding of the failure of certainty and interest/curiosity to mediate the relationship between anxious attachment orientation and epistemic trust.

It also leaves room for uncertainty about the poor mediation between avoidant attachment orientation and epistemic trust. Experimental and neuroimaging studies could offer deeper insights and validate the current findings. Including additional constructs could lead to a comprehensive model that integrates cognitive, emotional, and mental health aspects, enhancing our understanding of educational studies.

Although PROCESS was used as a pragmatic choice for estimating indirect effects with observed composite scores, this approach does not model latent variables or account for measurement error, limiting the generalizability and robustness of the findings. In the future more complex models need to be examined using composite-based SEM methods such as partial least squares (PLS-SEM).

Another significant limitation is the disproportionate number of female participants, stemming from a notable imbalance in favor of women in Schools of Social Sciences from which the participants were recruited. Previous studies have shown

gender differences in mentalizing and psychological mindedness among young adults [8]. The predominance of social studies students also restricts the generalizability of the study results to a more diverse student population, and to other academic disciplines. Using a clinical sample, such as students referred to a university counseling center, might reveal different associations among variables. Exploring secure attachment could offer a complementary perspective to the current study's findings. Further studies could take into account more diverse demographic variables to provide a more comprehensive view of emotional factors and mental health in higher education.

The cross-sectional design of this study limits the ability to track individual developmental changes over time. While comparing junior and senior students provides valuable insights, this approach cannot account for potential cohort effects or definitively attribute observed differences to developmental processes rather than other factors. A longitudinal design would be more appropriate for examining developmental changes in mentalizing, emotion regulation, and epistemic trust across university years.

Additionally, the internal consistency (reliability) values of the Interest/curiosity and Mistrust subscale were below the generally accepted threshold of 0.70, raising concerns about the reliability of the measurements for these constructs; that said, all findings related to these variables should be approached with caution.

Finally, examining mentalizing of the self and others separately might yield different results. Future research could further investigate the differences between junior and senior students in mentalizing, epistemic trust, learning, and adjustment constructs. This line of research could broaden the scope of studies focused on student profiles, often characterized by 'dissonance' in educational literature [2,4,87]. By considering the mentalizing system, future studies could reveal the contribution of mentalizing and ET to student learning and adjustment, providing valuable insights for educators and policymakers.

## Author contributions

**Conceptualization:** Evangelia Karagiannopoulou, Christos Rentzios, Peter Fonagy.

**Data curation:** Panagiotis Lianos, Christos Rentzios.

**Formal analysis:** Panagiotis Lianos.

**Funding acquisition:** Peter Fonagy.

**Investigation:** Evangelia Karagiannopoulou.

**Methodology:** Evangelia Karagiannopoulou, Christos Rentzios.

**Project administration:** Evangelia Karagiannopoulou, Peter Fonagy.

**Resources:** Panoraia Andriopoulou, Peter Fonagy.

**Supervision:** Evangelia Karagiannopoulou, Panoraia Andriopoulou, Peter Fonagy.

**Validation:** Evangelia Karagiannopoulou, Panoraia Andriopoulou.

**Visualization:** Panagiotis Lianos.

**Writing – original draft:** Evangelia Karagiannopoulou, Panagiotis Lianos, Panoraia Andriopoulou, Christos Rentzios, Peter Fonagy.

**Writing – review & editing:** Evangelia Karagiannopoulou, Panagiotis Lianos, Panoraia Andriopoulou, Christos Rentzios, Peter Fonagy.

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
