## [Decision Letter · Decision Letter 0]

PONE-D-24-19925The role of affect regulation and mentalizing in mediating the attachment-epistemic trust relationship. Differences between junior and senior students…Who is at risk?PLOS ONE?

Dear Dr. Lianos,

Thank you for submitting your manuscript to PLOS ONE. After careful consideration, we feel that it has merit but does not fully meet PLOS ONE’s publication criteria as it currently stands. Therefore, we invite you to submit a revised version of the manuscript that addresses the points raised during the review process.

Dear author/s,

Thank you for your submission. I have now received the reviews from the reviewers. After careful consideration, we feel that it has merit but needs revision. Please carefully address the comments, provide a line-by-line response letter, and highlight all the changes you make with different comments. If you disagree with the reviewers' comments, please write a rebuttal justifying why you disagree. Thank you

We look forward to receiving your revised manuscript.

Kind regards,

Ehsan Namaziandost

Academic Editor

PLOS ONE

Additional Editor Comments (if provided):

Reviewers' comments:

Reviewer's Responses to Questions

**Comments to the Author**

1. Is the manuscript technically sound, and do the data support the conclusions?

Reviewer #1: Yes

2. Has the statistical analysis been performed appropriately and rigorously?

Reviewer #1: Yes

3. Have the authors made all data underlying the findings in their manuscript fully available?

Reviewer #1: No

4. Is the manuscript presented in an intelligible fashion and written in standard English?

Reviewer #1: Yes

Reviewer #1: The manuscript makes a valuable contribution to the existing literature, particularly in the context of students' mental health, emotion regulation, and academic learning and achievement. However, there are several recommendations that should be taken into consideration.The title should be clearer, shorter, and more understandable. The three dots are not helpful in achieving clarity and are not in line with academic writing style. The abstract should provide a clearer explanation of the study's rationale, highlighting its relevance for student populations and briefly explaining its implications. The introduction is repetitive; please shorten the sections focusing on the main hypothesis, avoid redundancy in the hypothesis, and clearly define the variables based on the instruments used. Provide more details on the response rate. Provide information if there were any exclusion criteria. The sample has a gender imbalance, and this affects the generalizability of findings. Did the authors consider steps to reduce gender imbalance and recruit more male students? I recommend detailing which social media platforms were used. Why did you choose ECR-R as a measurement? It does not measure secure attachment. How could the sample be categorized into avoidant or anxious attachment styles? It is not logical to divide the sample into these two categories. How did you code the instrument? Please clarify and state how ECR-R measures attachment dimensions, and how secure attachment is not measured directly, but is inferred by low scores on avoidant and anxious attachment scores. Put this in the limitations of the study. Please give information on the validity of the instruments and specifically state the reliability of the instruments from the sample compared to that of the original scales. It is not usual to have junior and senior students at 72 years old. Please explain that. If the sample is part of a broader community, please explain. Please follow the journal guidelines for formatting tables. Briefly explain why the bootstrap method is used and briefly explain Zhao et al.'s approach for interpreting the mediation model. Please divide the result section into subsections, such as descriptive and correlations analysis, mediation analyses, as it is difficult to read. How is categorizing the correlations as mild, moderate, and large helpful for your hypothesis? Explain why you categorized the correlations and provide correlation coefficients in the writing of the results. In the result section, please just state the findings but do not interpret them. The discussion section is long and ideas are repetitive. It is difficult to read. Elaborate more on findings that are not consistent with the literature.

**Do you want your identity to be public for this peer review?** For information about this choice, including consent withdrawal, please see our Privacy Policy

Reviewer #1: No

---

## [Author Response · Author response to Decision Letter 1]

12 Jan 2025

Dear Editor and Reviewer, thank you for allowing us to revise and resubmit our manuscript and for your insightful comments and instructions to improve our manuscript. Please find our answers per comment as follows:

-Reviewer’s Comment: The manuscript makes a valuable contribution to the existing literature, particularly in the context of students' mental health, emotion regulation, and academic learning and achievement. However, several recommendations should be taken into consideration.

The title should be clearer, shorter, and more understandable. The three dots are not helpful in achieving clarity and are not in line with academic writing style.

Answer: We changed the title leaving out the dots and making it shorter

Attachment and epistemic trust in junior and senior university students. The mediating role of affect regulation and mentalizing. Who is at risk?

-Reviewer’s Comment: The abstract should provide a clearer explanation of the study's rationale, highlighting its relevance for student populations and briefly explaining its implications.

Answer: We made the following alterations to the Abstract to make it more readable.

Research on emotional factors and mental health in higher education has gained traction. Much attention has focused on first-year students as a potentially at-risk group, though some studies suggest that all students might face similar risks. This study examines differences between junior and senior undergraduates in terms of mentalizing, emotion regulation (ER), and psychological mindedness, involving cognitive capacities significantly developed by late adolescence. These constructs relate to understanding one's own and others' mental states, potentially mediating the relationship between attachment and epistemic trust (ET). The current study includes 460 undergraduate students, most of whom are female (96%). Results show that senior students score higher on reappraisal, certainty, and interest/curiosity compared to junior students. However, these factors did not mediate the relationship between anxious attachment orientation and ET. Certainty and interest/curiosity mediated the relationship between avoidant attachment orientation and ET, suggesting similar mediation patterns for junior and senior students. On the other hand, suppression and uncertainty/confusion were critical mediators in the relationship between insecure (anxious and avoidant) attachment orientations and epistemic trust. Findings suggest that universities should (a) foster environments that support psychological capacity and facilitate positive learning experiences, and (b) enhance epistemic trust through safe curiosity and develop protective and preventive interventions.

-Reviewer’s Comment: The introduction is repetitive; please shorten the sections focusing on the main hypothesis, avoid redundancy in the hypothesis, and clearly define the variables based on the instruments used.

Answer: This point has been remedied accordingly (p. 10).

Based on the Desatnik et al. study (2023), it is hypothesized that senior students will show higher levels of emotion regulation, psychological mindedness, and mentalizing compared to junior students (H1). Furthermore, junior students are expected to report the negative aspects of mentalizing (uncertainty/confusion) to mediate the relationship between insecure attachment and ET (H2), and senior students will report the positive aspects of mentalizing (certainty, interest/curiosity) to mediate the relationship between insecure attachment and ET (H3). Suppression is expected to mediate the relationship between avoidance attachment and ET for both junior and senior students (H4); reappraisal is hypothesized to mediate the relationship between anxious attachment and ET only for senior students. Suppression will mediate the relationship between anxious attachment and epistemic trust reported only by senior students (H5).

-Reviewer’s Comment: Provide more details on the response rate. Provide information if there were any exclusion criteria. The sample has a gender imbalance, and this affects the generalizability of findings.

Did the authors consider steps to reduce gender imbalance and recruit more male students?

Answer: The gender imbalance in the sample is expected since students from social and education sciences were mostly recruited (p.11). This limitation is mentioned in the Discussion section.

-Reviewer’s Comment: I recommend detailing which social media platforms were used.

Answer: We added a list of social media platforms we used (p. 11).

-Reviewer’s Comment: Why did you choose ECR-R as a measurement? It does not measure secure attachment. How could the sample be categorized into avoidant or anxious attachment styles? It is not logical to divide the sample into these two categories. How did you code the instrument? Please clarify and state how ECR-R measures attachment dimensions, and how secure attachment is not measured directly but is inferred by low scores on avoidant and anxious attachment scores. Put this in the limitations of the study.

Answer: The instrument chosen is the one most widely used to assess adult attachment. It is an instrument with good psychometric properties (Wei et al., 2007). It is indicative that the original article testing the psychometric properties of the short form of the scale has been cited by 1821 documents. The scale has been adapted into many languages with sound psychometric properties (e.g., Brenk-Franz et al., 2018; Imran et al., 2020). Adult attachment is predominately conceptualized as a dimensional construct since attachment security and attachment anxiety and avoidance (insecurity) exist along a continuum (Fraley et al., 2015). More details on the coding of the instrument have been added in the methods section (p. 12) and the limitations of using a dimensional measure have been mentioned in the discussion section (p. 28).

-Reviewer’s Comment: Please give information on the validity of the instruments and specifically state the reliability of the instruments from the sample compared to that of the original scales.

Answer: Further information about construct validity and original scales’ reliability was added to the text (p. 14-15). Specifically, all data are presented in a table (Table 3) so the reader can have a clear vision inspection of the relevant information.

-Reviewer’s Comment: It is not usual to have junior and senior students at 72 years old. Please explain that. If the sample is part of a broader community, please explain.

Answer: The case is exactly as mentioned in the comment. We proceeded to clarify this in the text by omitting redundant information and focusing on the present study (p.11).

-Reviewer’s Comment: Please follow the journal guidelines for formatting tables.

Answer: The required corrections were made according to the journal’s guidelines.

-Reviewer’s Comment: Briefly explain why the bootstrap method is used and briefly explain Zhao et al.'s approach for interpreting the mediation model.

Answer: We provided a brief explanation in the text for both procedures, bootstrapping and the interpretation of mediation models by Zhao et al. (2010).

-Reviewer’s Comment: Please divide the result section into subsections, such as descriptive and correlations analysis, and mediation analyses, as it is difficult to read.

Answer: We changed the titles of the subsections and this allowed us to re-arrange the text mostly of the assumptions section, making it more consistent and readable (p.17-18).

-Reviewer’s Comment: How is categorizing the correlations as mild, moderate, and large helpful for your hypothesis? Explain why you categorized the correlations and provide correlation coefficients in the writing of the results.

Answer: The sections of the data analyses concerning the correlations (p.14) and the findings (p. 15-17) were revisited to remedy this comment. We changed the nomenclature slightly to follow Guildford’s classification and added the appropriate references. Furthermore, the correlation coefficients were added to the text, making it easier for the reader to follow the results in Table 2.

-Reviewer’s Comment: In the result section, please just state the findings but do not interpret them.

Answer: We made some adjustments to omit sentences that could seem to explain the results.

-Reviewer’s Comment: The discussion section is long and ideas are repetitive. It is difficult to read. Elaborate more on findings that are not consistent with the literature.

Answer: We made some adjustments to omit sentences that were repetitive and make the text more concise. The excerpts that were crossed out can be seen in the Manuscript with Track Changes, especially in the beginning of the Discussion (p. 22) and in the Implementation section (p. 28).

---

## [Decision Letter · Decision Letter 1]

PONE-D-24-19925R1Attachment and epistemic trust in junior and senior university students. The mediating role of affect regulation and mentalizing. Who is at risk?PLOS ONE?

Dear Dr. Lianos,

Thank you for submitting your manuscript to PLOS ONE. After careful consideration, we feel that it has merit but does not fully meet PLOS ONE’s publication criteria as it currently stands. Therefore, we invite you to submit a revised version of the manuscript that addresses the points raised during the review process.

**Dear Author/s,**
**Thank you for submitting your manuscript to PLOS ONE. We appreciate the time and effort you have invested in this submission and your contribution to advancing research in this field.**
**After careful consideration and review, we regret to inform you that the reviewers have recommended major revisions to your manuscript. Their detailed comments and suggestions are attached to this email for your reference.**
**To move forward with the evaluation of your manuscript, we kindly ask that you consider the reviewers’ comments very carefully. Please make the necessary revisions and highlight all changes in the revised manuscript using a distinct color to ensure clarity. Additionally, we request that you provide a detailed, point-by-point response to the reviewers’ comments. In your response, please clearly outline how each comment has been addressed or provide a rationale if you believe no changes are needed.**
**To assist in the revision process, here are some key points to consider:**
**1. Ensure that the revised manuscript aligns with PLOS ONE's guidelines and standards.**
**2. Address all comments thoroughly and transparently.**
**3. Highlight any additional updates or clarifications you make to strengthen the manuscript.**
**Please submit your revised manuscript and the response letter through the online submission system by the due date, or inform us if you require additional time to complete the revisions.**
**We value your efforts and are committed to working with you to ensure your manuscript reaches its highest potential. If you have any questions or require clarification on the reviewers’ comments or the revision process, please do not hesitate to contact us.**
**Thank you for choosing PLOS ONE as a platform for your research. We look forward to receiving your revised manuscript.**
**Best regards,**
**Ehsan Namaziandost **
**Academic Editor **
**PLOS ONE Editorial Office**

We look forward to receiving your revised manuscript.

Kind regards,

Ehsan Namaziandost

Academic Editor

PLOS ONE

Reviewers' comments:

Reviewer's Responses to Questions

**Comments to the Author**

Reviewer #1: All comments have been addressed

2. Is the manuscript technically sound, and do the data support the conclusions?

Reviewer #1: Yes

3. Has the statistical analysis been performed appropriately and rigorously?

Reviewer #1: Yes

4. Have the authors made all data underlying the findings in their manuscript fully available?

Reviewer #1: Yes

5. Is the manuscript presented in an intelligible fashion and written in standard English?

Reviewer #1: Yes

**Reviewer #1: ** The authors have improved the manuscript and addressed most of the comments. However, there are some changes that are needed before the manuscript can be published. First, the authors report that the validity of the scales has been discussed elsewhere; however, no other information was provided. Simply stating that the validity of the scales has been discussed elsewhere is not transparent and does not provide at least a short description of the main report on the validity of the scales used in a different socio-cultural context. Second, the authors use PROCESS for the mediation analysis; however, the program has serious limitations compared to other programs such as MPlus. The rationale for using PROCESS instead of other programs should be included, and the limitations section should address this while recommending more advanced and rigorous models for future studies. Third, in the discussion section, the authors overgeneralize their findings by interpreting the results using brain maturation concepts, despite the fact that no neural processes were measured. These interpretations are speculative, and the authors should be cautious, primarily recommending further studies to measure neurological mechanisms in combination with self-report data. Overall, the results are relevant to the field of youth development and contribute to extending the literature on students' mental health and emotional well-being.

**Do you want your identity to be public for this peer review?** For information about this choice, including consent withdrawal, please see our Privacy Policy

Reviewer #1: No

---

## [Author Response · Author response to Decision Letter 2]

22 Mar 2025

Dear Editor and Reviewer, thank you for your kind remarks on our manuscript and for your insightful comments and instructions to improve our manuscript. Please find our answers per comment as follows:

-Reviewer’s Comment: First, the authors report that the validity of the scales has been discussed elsewhere; however, no other information was provided. Simply stating that the validity of the scales has been discussed elsewhere is not transparent and does not provide at least a short description of the main report on the validity of the scales used in a different socio-cultural context.

Answer: The sample of the present sample (students) stems from a larger community sample. Therefore, the measures used are the same and they have been reported in detail as to their psychometric properties in Karagiannopoulou et al. (2024). For clarity, we added the appropriate statements in each of the measures in the relevant section (pp.12-13) and a more specific sentence about convergent and structural validity in the Results section (p.14).

-Reviewer’s Comment: Second, the authors use PROCESS for the mediation analysis; however, the program has serious limitations compared to other programs such as MPlus. The rationale for using PROCESS instead of other programs should be included, and the limitations section should address this while recommending more advanced and rigorous models for future studies.

Answer: This comment holds merit and resonates the discussion about the advantages and disadvantages of the statistical procedures used to estimate mediations/moderations. Concerning the use of PROCESS, it has been widely used in mediational analyses as being relatively easy and straight-forward to use, especially for simple mediations as is the case of our study. Furthermore, the use of PROCESS with bootstrapping compared to the Sobel test allows for “re-sampling while requiring fewer assumptions, provides a higher study power, and lowers the risk of falsely rejecting the null hypothesis” (Abu-Bader & Jones, 2021, p.57). Furthermore, Hayes et al. (2017) argue that factor-based SEM methods are better performing using PROCESS compared with other programs such as AMOS, LISREL, EQS, and Mplus, especially when it comes to the generation of interaction terms or sample size requirements. Our analyses are performed with observed variables, and although for more composite models Partial Least Square-SEM seems more appropriate, PROCESS is more suitable for singular models running ordinary least squares regressions (Sarstedt et al., 2020). A relevant suggestion for future complex mediation analyses was added in the Limitations section.

-Reviewer’s Comment: Third, in the discussion section, the authors overgeneralize their findings by interpreting the results using brain maturation concepts, despite the fact that no neural processes were measured. These interpretations are speculative, and the authors should be cautious, primarily recommending further studies to measure neurological mechanisms in combination with self-report data.

Answer: In the Discussion it is added that neuroimaging studies have also reported a discrepancy between suppression and reappraisal, beyond the self-report studies (p. 27). Furthermore, in the Limitations section it is suggested that “Including additional constructs could lead to a comprehensive model that integrates cognitive, emotional, and mental health aspects, enhancing our understanding of educational studies” (p.29).

References:

Abu-Bader, S., & Jones, T. V. (2021). Statistical mediation analysis using the sobel test and hayes SPSS process macro. International Journal of Quantitative and Qualitative Research Methods, 9(1), 42-61.

Hayes, A. F., Montoya, A. K., & Rockwood, N. J. (2017). The analysis of mechanisms and their contingencies: PROCESS versus structural equation modeling. Australasian Marketing Journal, 25, 76–81.

Karagiannopoulou E, Milienos FS, Desatnik A, Rentzios C, Athanasopoulos V, Fonagy P. A short version of the reflective functioning questionnaire: Validation in a greek sample. PloS One. 2024; 19(2):e0298023. doi.org/10.1371/journal.pone.0298023

Sarstedt, M., Hair, J. F., Nitzl, C., Ringle, C. M., & Howard, M. C. (2020). Beyond a tandem analysis of SEM and PROCESS: Use of PLS-SEM for mediation analyses! International Journal of Market Research, 62(3), 288-299. https://doi.org/10.1177/1470785320915686

---

## [Decision Letter · Decision Letter 2]

PONE-D-24-19925R2Attachment and epistemic trust in junior and senior university students. The mediating role of affect regulation and mentalizing. Who is at risk?PLOS ONE?

Dear Dr. Lianos,

Thank you for submitting your manuscript to PLOS ONE. After careful consideration, we feel that it has merit but does not fully meet PLOS ONE’s publication criteria as it currently stands. Therefore, we invite you to submit a revised version of the manuscript that addresses the points raised during the review process.

**Dear authors,**
**The reviewers have acknowledged the strong potential of your paper but have highlighted a few minor issues that need to be addressed prior to final acceptance.**
**Best,**
**Dr. Ehsan Namaziandost**

We look forward to receiving your revised manuscript.

Kind regards,

Ehsan Namaziandost

Academic Editor

PLOS ONE

Journal Requirements:

Reviewers' comments:

Reviewer's Responses to Questions

**Comments to the Author**

Reviewer #1: All comments have been addressed

Reviewer #2: All comments have been addressed

2. Is the manuscript technically sound, and do the data support the conclusions?

Reviewer #1: Partly

Reviewer #2: Yes

3. Has the statistical analysis been performed appropriately and rigorously?

Reviewer #1: Yes

Reviewer #2: Yes

4. Have the authors made all data underlying the findings in their manuscript fully available?

Reviewer #1: Yes

Reviewer #2: Yes

5. Is the manuscript presented in an intelligible fashion and written in standard English?

Reviewer #1: Yes

Reviewer #2: Yes

Reviewer #1: I thank the authors and acknowledge their efforts to improve their manuscript. However, there are some further clarification that the authors should improve. First, the statement that PROCESS could better perform than MPLUS or SEM is misleading. PROCESS is effective in regression mediation analysis with observed variables but not with latent variables. Thus, PROCESS does not account for measurement error, does not provide a model fit in order to increase the possibility for generalizability and robustness of findings. This study uses self-report for psychological constructs, which are manifested as latent constructs, so claiming that is more suitable for observed variables is a key assumption violation. The literature that the authors provide is more focused on consumer behavior using observed metrics but not typically involving latent traits such as emotion regulation, mentalizing or attachment patterns. Please clarify this in the manuscript to be as clear as possible, in order for future studies to replicate with more advanced statistical models. Please review the citation in the paragraph of methodology: “Mediation hypotheses were tested using the PROCESS Procedure for SPSS…. (36 is not in line the citation with the reference) and then…. Analyses were performed with observed variables (not correct these are latent variables).

Second, the brain and neuroimaging literature does not specifically address the current study aims, as this study does not include any brain or neural data. Thus, referring to brain maturation processes to explain self-report results is speculative and go beyond the focus of your data. Please, revise each paragraph that have been addressing brain and neural interpretation to your data and remove them. Please, explicitly acknowledge that your interpretations based on neural development are hypothetical and require empirical validation in future research that directly measures neurobiological process, please use the information as a tentative explanation but that is unmeasured in the current study and that it should be interpreted with cautiousness. Even if these studies provide a theoretical rationale for grouping of students, however they should not be used to interpret the results of the study directly. Furthermore, referring to brain maturation process in the discussion section risks overstating the evidential basis of your conclusions, as the study does not support these results empirically. Be explicit in the statement that studies of brain and neural development cannot test these claims directly in your study, and in the discussion section state that this interpretations are speculative and that future studies could test empirically, and please in the limitations state clearly that the lack of brain and neural findings limits the conclusions about underlying brain or neural mechanisms, please remove terms like mentalizing system activation as this term implies neural activation systems which were not measured in the current study. Focus more on behavioral or psychological terms like mentalizing processes.

Reviewer #2: This study provides valuable insights into how emotional factors and mental health influence undergraduate students' academic experiences, with a particular focus on differences between third- and fourth-year students. The research's focus on mentalizing, emotion regulation, and psychological mindfulness provides a comprehensive framework for understanding how attachment styles influence students' emotional and cognitive processes. The large sample size (460 participants) and the clear differentiation between third- and fourth-year students contribute to the robustness of the findings. The study's suggestion that emotion regulation factors such as reappraisal and curiosity may mediate the relationship between attachment styles and epistemic trust represents an important and interesting addition to the field of educational psychology. This research could inform interventions to support students' mental health and emotional well-being, particularly in higher education institutions.

However, while the study offers interesting results, there are several areas for improvement. The predominantly female sample (96%) raises concerns about the generalizability of the findings to a more diverse student population. Additionally, the lack of a clear explanation of how the concepts of mindfulness and emotional regulation were measured may make it difficult for readers to fully understand the reliability and validity of the findings. The study's inability to demonstrate a mediating role for emotional regulation factors between anxious attachment and cognitive confidence raises questions about the robustness of the proposed theoretical framework. Furthermore, the article could benefit from a deeper exploration of how the findings could directly inform practical interventions in higher education, as the implications for teachers and counselors remain somewhat ambiguous. Finally, the study could have included more diverse demographic variables to provide a more comprehensive view of emotional factors and mental health in higher education.

make sure that the references follow the style specified by the journal.

**Do you want your identity to be public for this peer review?** For information about this choice, including consent withdrawal, please see our Privacy Policy

Reviewer #1: No

Reviewer #2: No

---

## [Author Response · Author response to Decision Letter 3]

31 May 2025

Dear Reviewers, we greatly appreciate the time and effort you have invested in reviewing our submission entitled “Attachment and epistemic trust in junior and senior university students. The mediating role of affect regulation and mentalizing. Who is at risk?”. We would like to thank you for your thoughtful and constructive comments on our manuscript. We have carefully considered all the comments and have revised the manuscript accordingly. Below you can find detailed responses to each comment:

Reviewer #1: I thank the authors and acknowledge their efforts to improve their manuscript. However, there are some further clarification that the authors should improve.

-Comment: First, the statement that PROCESS could better perform than MPLUS or SEM is misleading. PROCESS is effective in regression mediation analysis with observed variables but not with latent variables. Thus, PROCESS does not account for measurement error, does not provide a model fit in order to increase the possibility for generalizability and robustness of findings. This study uses self-report for psychological constructs, which are manifested as latent constructs, so claiming that is more suitable for observed variables is a key assumption violation. The literature that the authors provide is more focused on consumer behavior using observed metrics but not typically involving latent traits such as emotion regulation, mentalizing or attachment patterns. Please clarify this in the manuscript to be as clear as possible, in order for future studies to replicate with more advanced statistical models. Please review the citation in the paragraph of methodology: “Mediation hypotheses were tested using the PROCESS Procedure for SPSS…. (36 is not in line the citation with the reference) and then…. Analyses were performed with observed variables (not correct these are latent variables).

Answer: We thank the reviewer for this insightful and important comment. We acknowledge that our original statement may have been misleading in its comparison between PROCESS and more advanced SEM tools such as Mplus. As correctly pointed out, PROCESS is limited to mediation analyses using observed variables and does not account for measurement error, nor does it provide overall model fit indices. Given that our study involves self-reported measures of psychological constructs, which are best conceptualized as latent variables (e.g., emotion regulation, mentalizing, and attachment), the use of PROCESS indeed comes with methodological limitations. We have revised the manuscript accordingly to clarify that PROCESS was used as a pragmatic choice for estimating indirect effects with observed composite scores, but that this approach does not model latent variables or account for measurement error. We now explicitly state that this limits the generalizability and robustness of our findings and that future research should ideally replicate these findings using latent variable modeling via SEM approaches. Additionally, we have corrected the misstatement regarding "analyses being performed on observed variables" and have revised the language to reflect that composite scores were used as proxies for latent constructs. The incorrect citation [36] in the methodology section has also been corrected and now aligns with the reference list [63]. Finally, we omitted the source that previously corresponded to [89] and replaced it with a more suitable that describes the PROCESS approach by Hayes.

“…analytical framework to evaluate both direct and indirect effects with observed composite scores as proxies for latent variables [63]. All outcome variables were continuous, and the standard assumptions of ordinary least squares regression (normality, independence, and homoscedasticity) were tested [89].”

“Although PROCESS was used as a pragmatic choice for estimating indirect effects with observed composite scores, this approach does not model latent variables or account for measurement error, limiting the generalizability and robustness of the findings. In the future more complex models need to be examined using composite-based SEM methods such as partial least squares (PLS-SEM). ”

-Comment: Second, the brain and neuroimaging literature does not specifically address the current study aims, as this study does not include any brain or neural data. Thus, referring to brain maturation processes to explain self-report results is speculative and go beyond the focus of your data. Please, revise each paragraph that have been addressing brain and neural interpretation to your data and remove them. Please, explicitly acknowledge that your interpretations based on neural development are hypothetical and require empirical validation in future research that directly measures neurobiological process, please use the information as a tentative explanation but that is unmeasured in the current study and that it should be interpreted with cautiousness. Even if these studies provide a theoretical rationale for grouping of students, however they should not be used to interpret the results of the study directly. Furthermore, referring to brain maturation process in the discussion section risks overstating the evidential basis of your conclusions, as the study does not support these results empirically. Be explicit in the statement that studies of brain and neural development cannot test these claims directly in your study, and in the discussion section state that this interpretations are speculative and that future studies could test empirically, and please in the limitations state clearly that the lack of brain and neural findings limits the conclusions about underlying brain or neural mechanisms, please remove terms like mentalizing system activation as this term implies neural activation systems which were not measured in the current study. Focus more on behavioral or psychological terms like mentalizing processes.

Answer: In the text all neurological terminology was removed or replaced with phrases that imply psychological processes (instead of systems). We also added that interpretations tapping into neurological research were not based on the study’s findings and need to be further empirically tested. Please find below the respectful excerpts:

“From this perspective, mentalizing abilities and attachment relationships are seen as loosely interconnected [18; 19]. Moreover, empirical evidence strongly supports the idea that both the attachment and mentalizing processes play central roles in stress and emotion regulation [20].”

“Research suggests that mentalizing tends to improve with age, with evidence showing changes throughout adolescence and into early adulthood. Particularly, Desatnik et al., (2023) suggest that mentalizing ability increases gradually from early adolescence to young adulthood and keeps mature beyond it, aligning with brain development. Notably, females aged 17-18 showed different mentalizing patterns compared to those over 20 [8]. In this line of thinking, Poznyak et al., (2025) demonstrate that while basic mentalizing processes exist in adolescence, as people get older, they become more flexible in how they shift between the “self-other” representations [90].”

“Despite growing interest in the relationship between attachment and student learning and adjustment, research on the way mentalizing operates remains limited.”

“These changes are part of the broader emotional development process; studies show that the cognitive control aspects of reappraisal become more active with age [44].”

“In the context of these processes, trust in others as sources of social information fosters resilience through a salutogenic process, allowing individuals to gain from the stream of relevant information in the social environment [20].”

“Our perspective that exploring the attachment and mentalizing processes can inform understandings of learning, success, and adjustment in higher education aligns with Moreau et al.'s [13] caution against "overstating the environmental factors in success" compared to individual traits.”

“For junior students, interest/curiosity reduced the positive association between attachment avoidance and mistrust. This "beneficial" mediation may reflect developmental limitations that could be cautiously – given the self-report methodology of the present study – linked to research showing that brain regions responsible for understanding mental states remain immature until late adolescence [17].”

“Moreover, excessive certainty enhanced the negative relationship between avoidant attachment and epistemic trust reported by seniors, possibly suggesting that the development of more complex “self-other” processes might raise stress levels and activate negative schemata regarding others, thus triggering defensive independence. Therefore, cognitive maturation possibly inhibits controlled mentalizing [17; 22].”

“The mediation by a construct that does not increase with age may indicate the preexisting issues associated with anxious attachment, such as a low threshold for rapid and intense activation of the attachment orientation and corresponding deactivation of controlled mentalizing processes, leading to automatic mentalizing [17].”

“This aligns with the attachment literature, which indicates that individuals with anxious attachment exhibit a rapid and extensive surge of negative emotions [75].”

“This vulnerability can be seen – although tendatively in the context of the present study – in combination with the still-developing mentalizing brain areas, to suggest a high likelihood of automatic rather than controlled mentalizing, reinforcing the idea that university students could be considered at risk.”

We also removed a reference to a neurological study [24] Sebastian CL, Fontaine NMG, Bird G, Blakemore SJ, De brito SA, Mccrory EJP, et al. Neural processing associated with cognitive and affective theory of mind in adolescents and adults. Social Cognitive and Affective Neuroscience 2012; 7:53–63. https://doi.org/10.1093/scan/nsr023 PMID: 21467048 replacing with “Freda, M. F., Esposito, G., & Quaranta, T. (2015). Promoting Mentalization in Clinical Psychology at Universities: A Linguistic Analysis of Student Accounts. Europe’s Journal of Psychology. 2015;11(1): 34–49. doi.org/10.5964/ejop.v11i1.812” that is purely psychological.

Reviewer #2: This study provides valuable insights into how emotional factors and mental health influence undergraduate students' academic experiences, with a particular focus on differences between third- and fourth-year students. The research's focus on mentalizing, emotion regulation, and psychological mindfulness provides a comprehensive framework for understanding how attachment styles influence students' emotional and cognitive processes. The large sample size (460 participants) and the clear differentiation between third- and fourth-year students contribute to the robustness of the findings. The study's suggestion that emotion regulation factors such as reappraisal and curiosity may mediate the relationship between attachment styles and epistemic trust represents an important and interesting addition to the field of educational psychology. This research could inform interventions to support students' mental health and emotional well-being, particularly in higher education institutions.

-Comment: However, while the study offers interesting results, there are several areas for improvement. The predominantly female sample (96%) raises concerns about the generalizability of the findings to a more diverse student population.

Answer: Thank you for this comment. This concern has been addressed in the Limitations section in the previous version:

“Another significant limitation is the disproportionate number of female participants, stemming from a notable imbalance in favor of women in Schools of Social Sciences from which the participants were recruited. Previous studies have shown gender differences in mentalizing and psychological mindedness among young adults [8]. The predominance of social studies students also restricts the generalizability of the study results to other academic disciplines”.

However, the clarification “to a more diverse student population” has been added in the text.

“The predominance of social studies students also restricts the generalizability of the study results to a more diverse student population, and to other academic disciplines. Using a clinical sample, such as students referred to a university counseling center, might reveal different associations among variables. Exploring secure attachment could offer a complementary perspective to the current study's findings.”

-Comment: Additionally, the lack of a clear explanation of how the concepts of (Psychological mindfulness) and emotional regulation were measured may make it difficult for readers to fully understand the reliability and validity of the findings.

Answer: The concepts of Psychological Mindedness (not mindfulness) and emotional regulation were measured with self-report instruments clearly described in the Measures section of the manuscript. Furthermore, psychometric properties of these measures have been reported in Karagiannopoulou et al. (2024) as stated in the manuscript: “Since the data were taken from a greater study, convergent and structural validity of the Greek versions of the original scales have been discussed elsewhere and have been found to have satisfactory psychometric properties [58]”.

-Comment: The study's inability to demonstrate a mediating role for emotional regulation factors between anxious attachment and cognitive confidence raises questions about the robustness of the proposed theoretical framework.

Answer: The results support previous studies and well-established findings that draw for the work of both Mikulincer & Shaver (2016) and Bateman & Fonagy (2019). The activation of the attachment system in insecure attached individuals, particularly those reporting anxious attachment, and the automatic rather than controlled mentalizing indicating the dominance of the attachment system, supports the poor mediating role of emotional regulation variables explored in the study.

-Comment: Furthermore, the article could benefit from a deeper exploration of how the findings could directly inform practical interventions in higher education, as the implications for teachers and counselors remain somewhat ambiguous.

Answer: Practical interventions in higher education aligned with the study’s findings have been added into the Discussion section

“Considering mentalizing as not just an individual but a social process, universities can create networks of relationships that facilitate thinking and appropriate action through mirroring, particularly in stressful situations, which in turn supports learning and affiliation [17;80]. Furthermore, findings suggest that interventions aimed at enhancing mentalization should be tailored developmentally, especially in early-to-mid adolescence [8]. Mentalizing can inform practical interventions in higher education by fostering university students’ reflective abilities through structured group settings, where they could explore interpersonal dynamics and emotional experiences in real time. Moreover, students through narrative writing tasks with guided reflection could enhance the development of self-other awareness and metacognitive skills [24].”

-Comment: Finally, the study could have included more diverse demographic variables to provide a more comprehensive view of emotional factors and mental health in higher education.

Answer: According to the comment, the following sentence was added into the Limitations section “Further studies could take into account more diverse demographic variables to provide a more comprehensive view of emotional factors and mental health in higher education.”

-Comment: Make sure that the references follow the style specified by the journal.

Answer: All necessary adjustments have been made so that references adhere to the journal’s specifications.

We hope that the changes we have made have addressed your concerns and improved the clarity and quality of the manuscript. We thank you again for the opportunity to revise our manuscript in a way that would fit best in the publication list of the journal.

---

## [Editor Report · Decision Letter 3]

Attachment and epistemic trust in junior and senior university students. The mediating role of affect regulation and mentalizing. Who is at risk?

PONE-D-24-19925R3

Dear Dr. Lianos,

We’re pleased to inform you that your manuscript has been judged scientifically suitable for publication and will be formally accepted for publication once it meets all outstanding technical requirements.

Kind regards,

Ehsan Namaziandost

Academic Editor

PLOS ONE
---

## [Editor Report · Acceptance letter]

PONE-D-24-19925R3

PLOS ONE

Dear Dr. Lianos,

I'm pleased to inform you that your manuscript has been deemed suitable for publication in PLOS ONE. Congratulations! Your manuscript is now being handed over to our production team.

Kind regards,

on behalf of

Dr. Ehsan Namaziandost

Academic Editor

PLOS ONE